# Intercomparison of Machine-Learning Methods for Estimating Surface Shortwave and Photosynthetically Active Radiation

**Meredith G. L. Brown** [1,*], **Sergii Skakun** [1], **Tao He** [2] **and Shunlin Liang** [1]

[1] Department of Geographical Sciences, University of Maryland, College Park, MD 20742, USA; skakun@umd.edu (S.S.); sliang@umd.edu (S.L.)

[2] School of Remote Sensing and Information Engineering, Wuhan University, Wuhan 430079, China; taohers@whu.edu.cn

[*] Correspondence: mglbrown@terpmail.umd.edu

**Abstract:** Satellite-derived estimates of downward surface shortwave radiation (SSR) and photosynthetically active radiation (PAR) are a part of the surface radiation budget, an essential climate variable (ECV) required by climate and vegetation models. Ground measurements are insufficient for generating long-term, global measurements of surface radiation, primarily due to spatial limitations; however, remotely sensed Earth observations offer freely available, multi-day, global coverage of radiance that can be used to derive SSR and PAR estimates. Satellite-derived SSR and PAR estimates are generated by computing the radiative transfer inversion of top-of-atmosphere (TOA) measurements, and require ancillary data on the atmospheric condition. To reduce computational costs, often the radiative transfer calculations are done offline and large look-up tables (LUTs) are generated to derive estimates more quickly. Recently studies have begun exploring the use of machine-learning techniques, such as neural networks, to try to improve computational efficiency. Here, nine machine-learning methods were tested to model SSR and PAR using minimal input data from the Moderate Resolution Imaging Spectrometer (MODIS) observations at 1 km spatial resolution. The aim was to reduce the input data requirements to create the most robust model possible. The bootstrap aggregated decision tree (Bagged Tree), Gaussian Process Regression, and Neural Network yielded the best results with minimal training data requirements: an $R^2$ of 0.77, 0.78, and 0.78 respectively, a bias of $0 \pm 6$, $0 \pm 6$, and $0 \pm 5$ W/m$^2$, and an RMSE of $140 \pm 7$, $135 \pm 8$, and $138 \pm 7$ W/m$^2$, respectively, for all-sky condition total surface shortwave radiation and viewing angles less than 55°. Viewing angles above 55° were excluded because the residual analysis showed exponential error growth above 55°. A simple, robust model for estimating SSR and PAR using machine-learning methods is useful for a variety of climate system studies. Future studies may focus on developing high temporal resolution direct and diffuse estimates of SSR and PAR as most current models estimate only total SSR or PAR.

**Keywords:** PAR; machine-learning; MODIS; shortwave radiation; radiative transfer; surface radiation; satellite remote sensing; radiation budget

## 1. Introduction

Earth's weather and climate are driven by the energy received at the surface, which is mainly affected by the contents of the atmosphere, including clouds and aerosols [1,2]. Despite the importance of downward surface radiation fluxes, studies have shown they are not always well estimated from satellite observations nor modeled by climate models [3], and the number of ground sites measuring radiation fluxes is itself a variable number. Surface stations can be difficult or expensive to maintain,

and there are many regions on Earth's surface which are virtually inaccessible and therefore are especially poorly represented by ground-measured data.

The observed trends in surface shortwave radiation (SSR) are not explained by changes in solar luminosity [1], therefore they must be due to changes in atmospheric conditions, including cloudiness, aerosols and greenhouse gases, which suggests that anthropogenic air pollution (aerosols) play a significant role in global dimming and brightening [4–6]. The observed trends in SSR fit with regional emission history [1].

Vegetation activity (photosynthesis) requires sunlight, precipitation, and favorable temperatures [7–9]. Wild [1] showed that SSR trends are associated with increasing trends in both precipitation and near-surface air temperature, while decreasing SSR trends are associated with decreasing precipitation in the northern hemisphere. Studies have shown, however, that in the northern hemisphere the increased fraction of diffuse radiation associated with global dimming increased the amount of carbon removed from the atmosphere during photosynthesis, suggesting that during the period of global dimming vegetation activity increased [10–13].

Current satellite-based estimates of surface radiation incorporate atmospheric information in their algorithms, which can be difficult to obtain and propagate error and uncertainty through the algorithm. A popular method for reducing the computational demands of generating a product is to compute the radiative transfer inversions offline and store them in a look-up table (LUT) [14–16]. LUTs can be generated using in situ data or simulated data, and their major advantage is the ability to do the radiative transfer inversion calculations ahead of time to speed up data generation [15]. The major disadvantage is that a LUT must be segmented into bins of a pre-defined size, and then estimates are interpolated between the values in the LUT. The finer the bin segments the larger the LUT, the longer it takes to generate the LUT, and the more time it takes to search the LUT to generate the data of interest. Balancing these requirements is the art of the LUT method. Other methods can also be used to optimize or parameterize a LUT [15], and the computational requirements of these methods is also a limitation of the overall LUT approach.

The aim of this study is to determine if it is reasonable to develop a machine-learning-based model for estimating SSR and PAR from TOA measurements alone. Traditionally surface radiation estimates are generated using physical-based, radiative transfer models [17,18]. These models typically require information about the top-of-atmosphere, the atmosphere, and the surface, or they can be parameterized to reduce the ancillary data requirements [19]. Acquiring this ancillary data introduces sources of potential error, requires heavy-duty computing resources, and is still time intensive [15]. Therefore, the goal of this study is to build an empirical model that only requires TOA data as input to reduce these extra sources of potential error, which can be trained and executed quickly and efficiently, while still yielding comparable results to existing methods. We have chosen to test a selection of machine-learning methods [20–22] in our model to explore how much of the physical processes they can capture as well as possibly improve on computational demands by selecting the smallest reasonable training samples.

Certain methods, such as linear regression methods and decision tree methods are simple and versatile, and have minimal parameters to adjust [20]. However, they work best for applications with large training samples available and data that fits certain assumptions about linear or Gaussian distributions. Other methods, like neural networks, have many more model parameters available for tuning to specific problems; however they are often criticized for being largely "black box", especially compared to linear methods and decision tree-based methods. Neural networks can often yield highly accurate results, but like linear methods and decision tree-based methods, they require large training samples. For applications where large training samples are not available, other methods like Kernel Ridge Regression and Gaussian Process Regression are available. These methods have a variety of hyperparameters to tune, but their main strength is in being able to make use of a smaller training sample during model development [20,22,23].

For this study, all selected methods are tested with minimal tuning and the best results will be identified for further study and development. Here, we use only MODIS TOA measurements and cloud condition, but the model could potentially be adapted to use higher spatial resolution observations such as the Harmonized Landsat Sentinel-2 data (HLS) [24] or they could be adapted for VIIRS [25,26] to extend the existing MODIS data record and incorporate further atmospheric or surface information.

## 2. Data

The data sources and years available are shown in Table 1. For the first part of the study, the initial intercomparison between machine-learning methods, the surface shortwave radiation (SSR) and photosynthetically active (PAR) models were trained using data from 2005–2009 and independently validated against data from 2010. In the second part of the study, the temporal stability test of the different machine-learning methods in the models, a Leave One Year Out Cross-Validation approach was used, described in Section 3.3. All ground truth data are from the SURFRAD sites located in the contiguous United States. Each year of data contains approximately 8200 combined satellite overpasses.

**Table 1.** Data used for model training and validation.

| Data | Years Avail. | Spatial Res. | Temporal Res. | Citation |
|---|---|---|---|---|
| MOD/MYD021KM TOA Reflectance | 2002–current | 1 km at nadir | instantaneous 1–2-day revisit | [27,28] |
| MOD/MYD35 Cloud Mask | 2002–current | 1 km at nadir | instantaneous 1–2-day revisit | [29] |
| MOD/MYD03 Geolocation | 2002–current | 1 km at nadir | instantaneous 1–2-day revisit | [30,31] |
| SURFRAD | 2003–current | 10 m footprint | 3-min before 2005 1-min since 2005 | [32] |

### 2.1. Remote Sensing

The model inputs are collected from the MODIS top-of-atmosphere (TOA) reflectance from both Terra and Aqua, MOD021KM and MYD021KM respectively, at 1 km spatial resolution. We use the reflectance of the first seven bands: red (620–670 nm), near Infrared (841–876 nm), blue (459–479 nm), green (545–565 nm), and the three shortwave infrared bands 1230–1250 nm, 1628–1652 nm, and 2105–2155 nm. Additional inputs to the SSR and PAR models are the satellite viewing geometry: solar zenith angle, satellite view zenith, and the relative angle between the solar and satellite azimuth (relative azimuth angle). We also use the cloud mask (MOD35 and MYD35) as a categorical variable to obtain the cloud condition since no other atmospheric information is explicitly contained in the models.

### 2.2. SURFRAD

The SSR and PAR models are trained and validated using the seven SURFRAD sites in the contiguous United States. The Surface Radiation Budget Network (SURFRAD) consists of seven ground sites in the United States [32] shown in Figure 1. The seven SURFRAD sites, which were all installed by 2003, allow for continuous monitoring of direct and diffuse total radiation and PAR at sites in different climate zones, with varying surface types and elevations. The sites have been maintained and updated since their installation, the data is provided in a consistent form with notifications about adjustments and errors to users. While there are other ground sites in the US and other countries as part of other networks, not all of them meet the same standard as the SURFRAD sites, and many were set up as part of short term experiments, and therefore do not have very long data records or the necessary variables available.

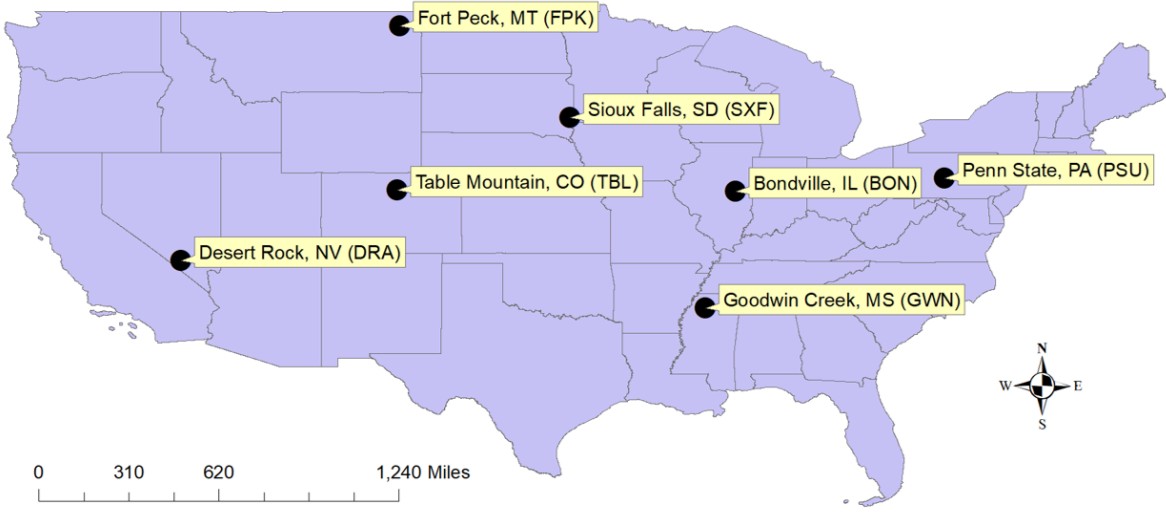

**Figure 1.** Map of the seven SURFRAD sites in the conterminous United States (CONUS).

The SURFRAD instruments, mounted on platforms 1.5 to 2 m off the ground, and the measurements we used for this experiment are: direct and diffuse solar radiation, and PAR. The direct radiation is measured with a normal incidence pyrheliometer (NIP) which is mounted on a sun tracker, while the diffuse radiation is measured with a shaded pyranometer also attached to the sun tracker. Using the direct and diffuse solar radiation measurements, total SSR is calculated as follows:

$$SSR = R_{dir} * \cos SZ + R_{dif} \tag{1}$$

where $R_{dir}$ is the direct component of radiation, $R_{dif}$ is the diffuse component, and SZ is the solar zenith angle. The uncertainty requirements for the SURFRAD instruments are 2–5% or 15 W/m$^2$ whichever is larger, to meet the World Climate Research Program specifications [33].

*2.3. Training and Validation Data Sets*

Prior to training, TOA reflectance from MOD021KM and MYD021KM and the overpass times are extracted for pixels containing the location of surface sites. We take a ±15 min temporal average of the SURFRAD data for each satellite overpass. For each site, only one pixel is selected, and no spatial averaging is done at this time following the methods of Zhang et al. [15] and Carter and Liang [34]. Over all sites and all years of data, we have a total of 51,142 data pairs.

**3. Methods**

In this study, four "families" of methods were tested, namely linear methods, decision tree methods, Neural Network-based methods, and kernel-based methods. Below is a brief overview of these types of methods.

*3.1. Modeling SSR and PAR with Machine-Learning Methods*

3.1.1. Linear Methods

Regularized Linear Regression [23] is used as the benchmark method here because it is the simplest, most straightforward method we can use, and one of the most transparent as it gives the most information about the relative importance of the input variables on the model output.

Two additional linear methods were tested, Least Absolute Shrinkage and Selection Operator (LASSO) [35,36] and Elastic Net Regularization (ELASTIC NET) [37,38]. The LASSO method is a type of feature selection and regularization method, which in its simplest form is a type of least squares regression model. ELASTIC NET is a method which includes both the feature selection and

regularization of the LASSO method, as well as ridge regression, both methods are supposed to improve prediction accuracy, especially for ill-posed problems.

### 3.1.2. Decision Tree Methods

Decision Trees are a type of non-parametric, supervised learning methods. They approximate a function by incrementally creating a set of if-then-else rules while breaking data sets into increasingly smaller subsets. A Bootstrap Aggregated (Bagged) Decision Tree [39] is a special case of the ensemble approach applied to the basic decision tree method that can reduce variance and avoid overfitting. The method works by sampling the original training set for each new tree to create an ensemble of trees from which predictions can be made.

### 3.1.3. Neural Networks

The feed-forward multi-layer perceptron (MLP) is one of the most common neural networks. In this method, the inputs are fed through the hidden layers and connected to the outputs through a series of weights. The outputs of each layer are compared to the desired outputs and fed back through the network, adjusting the weights each time, until the error function has been minimized [20,23,40,41].

### 3.1.4. Kernel Methods

Of the many different types of kernel methods, Gaussian Process Regression (GPR) [21,23] sometimes also known as kriging, is a type of distance weighting machine-learning algorithm that makes use of an assumed Gaussian probability distribution to make its predictions. This feature of the method requires small training sample sizes lest the model become too cumbersome.

### *3.2. Data Filtering, Parameter Tuning, and Training*

Our aim is to create an all-sky model, therefore we include all sky conditions identified according to the MODIS cloud mask. The inputs to the model are solar zenith angle (SZ), view zenith angle (VZ), relative azimuth (AZ), reflectance in the first seven top-of-atmosphere (TOA) MODIS bands, and the coded cloud condition described in Section 2.3.

Training data is filtered so that the training set contains only pixels whose cloud flag matches the expected amount of ground-measured radiation are used for training; however all valid pixels are included in the validation data, so there may be mismatches between the satellite cloud mask and the ground observed cloud condition.

Data viewed above 55° is discarded. Due to the bowtie effect, the pixel size at such extreme viewing angles is much larger than the pixel size at and nearer to nadir [42]. Furthermore, the additional path length through the atmosphere at such extreme viewing angles contaminates the pixel compared to small viewing angles. The MODIS team recommends against using pixels at such high viewing angles due to data quality issues associated with the extremity of the viewing angle [27].

Each method has a different optimal training sample size, for example, a Neural Network benefits from the largest training sample available, whereas a Gaussian Process Regression is optimized for small training sample sizes (1000–2000 points), therefore for each method, the model is allowed to separate its training and internal testing samples randomly, according to its optimal parameters. For this reason, several tests are done leaving specific data out for independent cross-validation and intercomparison.

Each method also has a different set of tunable parameters. The linear methods have at most one or two parameters to tune, whereas the Neural Network has several, including number of hidden layers, number of neurons per layer, number of epochs, and the Kernel Ridge Regression and Gaussian Process Regression have a gamma parameter that defines the kernel. For each combination of tunable parameters, tests were performed to find the final combination that yielded the best results with the smallest computational requirements.

During parameter tuning tests, the final combination of parameters selected had to maximize $R^2$, while minimizing RMSE and training time. The Bagged Tree reaches its peak performing parameters with only 200 trees in the bag, yet we tested up to 2000 trees to see if we could get any improvement on RMSE. It is possible that with more than 200 bags in the tree, the method overfits, other works suggest that the number of trees in the bag should depend on the number of features in the model [43–45]. In our model we have 14 input variables, thus it should be expected that a relatively small number of trees should be optimal.

For the final parameter selection, we used a Neural Network with 1 hidden layer, 14 neurons (one for each input), 100 training epochs, and is trained with the Levenberg-Marquardt backpropagation method. The Bagged Tree and Boosted Tree methods use 200 trees/bag. The Kernel Ridge Regression and Gaussian Process Regression use a radial basis (RBF) kernel. Other kernel functions tested were not successful.

The RLR method coefficients, shown in Figure 2, show that the MODIS red band (620–670 nm), where the peak solar energy is measured, and green band (545–565 nm), are the most influential input variables, and further that as reflectance in the blue (459–479 nm), green, and SWIR (1230–1250 nm) bands increases, the estimated SSR or PAR decreases, showing that these bands are the most sensitive to aerosols in the atmosphere that might lead to the global dimming phenomenon [1].

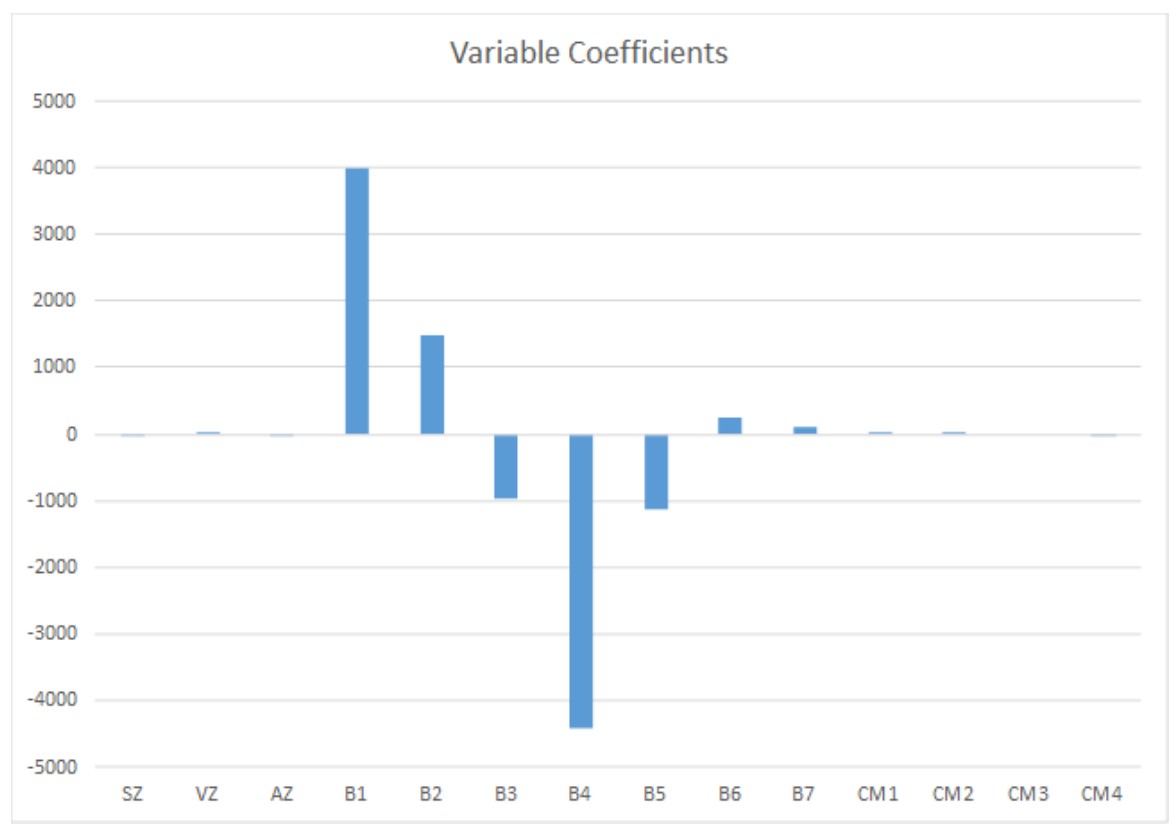

**Figure 2.** Relative importance of the model input variables.

### 3.3. Cross Validation

Each machine-learning method has an optimal training sample size. Linear regression and neural networks perform best with large training samples, whereas Kernel Ridge Regression (KRR) and Gaussian Process Regression (GPR) are better suited for smaller training data sets. In order to make the best use of each method we use different training samples sizes according to the method and then use three different methods for validation and intercomparison.

Validation Method (1): Data from 2005–2009 are used for the training set, while data from 2010 is used for independent validation and intercomparison. Using this training and validation method, we have 42,754 data pairs available for training and 8388 for validation.

Validation Method (2): We use the Leave One Year Out Cross-Validation (LOYOCV) method, a type of k-fold cross-validation, where we hold one year out and repeat the training and cross-validation 6 times, each time using five years for training and holding one year out for validation. On average, there are 1217 data pairs per site per year, meaning each iteration has an average of 42,618 data pairs available for training and 8523 for validation.

Validation Method (3): We use the Leave One Station Out Cross-Validation (LOSOCV) method, a similar type of k-fold cross-validation, and train on six of the seven SURFRAD stations and hold one out for cross-validation, iterating through just as we did for the LOYOCV. On average, each iteration in this validation method has 43,836 data pairs available for training and 7306 for validation. By using this type of cross-validation, we build an ensemble from which we can determine the spatial and temporal stability. In order to evaluate the different machine-learning methods and compare them to each other we calculate $R^2$, RMSE, and bias as follows:

$$R^2 = \frac{\sum \left( R_{est} - \overline{R_{obs}} \right)^2}{\sum \left( R_{obs} - \overline{R_{obs}} \right)^2} \tag{2}$$

$$RMSE = \sqrt{\frac{\sum \left( R_{est} - R_{obs} \right)^2}{n}} \tag{3}$$

$$Bias = \frac{\sum \left( R_{est} - R_{obs} \right)}{n} \tag{4}$$

where $R_{est}$ are the modeled surface radiation (either SSR or PAR), $R_{obs}$ are the ground-measured radiation data, $\overline{R_{obs}}$ is the mean of the ground-measured radiation, and $n$ are the number of data pairs.

## 4. Results

### 4.1. Model Performance

The results of the nine machine-learning methods, shown in Table 2 show that the linear methods and a single decision tree do not simulate the ground observed SSR and PAR as well as the non-linear methods. The best methods for SSR and PAR are the bootstrap aggregated (BAGGED TREE) decision tree, the single hidden layer Neural Network (NN), and the Gaussian Process Regression (GPR) methods and are shown in Figure 3.

**Table 2.** Validation Method 1 results for both SSR and PAR.

| Method | SSR $R^2$ | SSR RMSE (W/m$^2$) | SSR Bias (W/m$^2$) | PAR $R^2$ | PAR RMSE (W/m$^2$) | PAR Bias (W/m$^2$) |
|---|---|---|---|---|---|---|
| RLR | 0.68 | 170 (29%) | −11 | 0.70 | 69 (29%) | −3 |
| LASSO | 0.68 | 170 (29%) | 28 | 0.70 | 70 (29%) | 11 |
| ELASTIC NET | 0.69 | 170 (29%) | 29 | 0.70 | 70 (29%) | 11 |
| DECISION TREE | 0.62 | 190 (31%) | −8 | 0.60 | 82 (31%) | −3 |
| BAGGED TREE | 0.77 | 144 (23%) | −8 | 0.76 | 61 (23%) | −2 |
| BOOSTED TREE | 0.73 | 155 (25%) | −11 | 0.73 | 65 (24%) | −3 |
| NN | 0.78 | 138 (22%) | −4 | 0.78 | 59 (22%) | −1 |
| KRR | 0.75 | 149 (24%) | −7 | 0.75 | 62 (23%) | −1 |
| GPR | 0.78 | 140 (23%) | −5 | 0.78 | 59 (22%) | −2 |

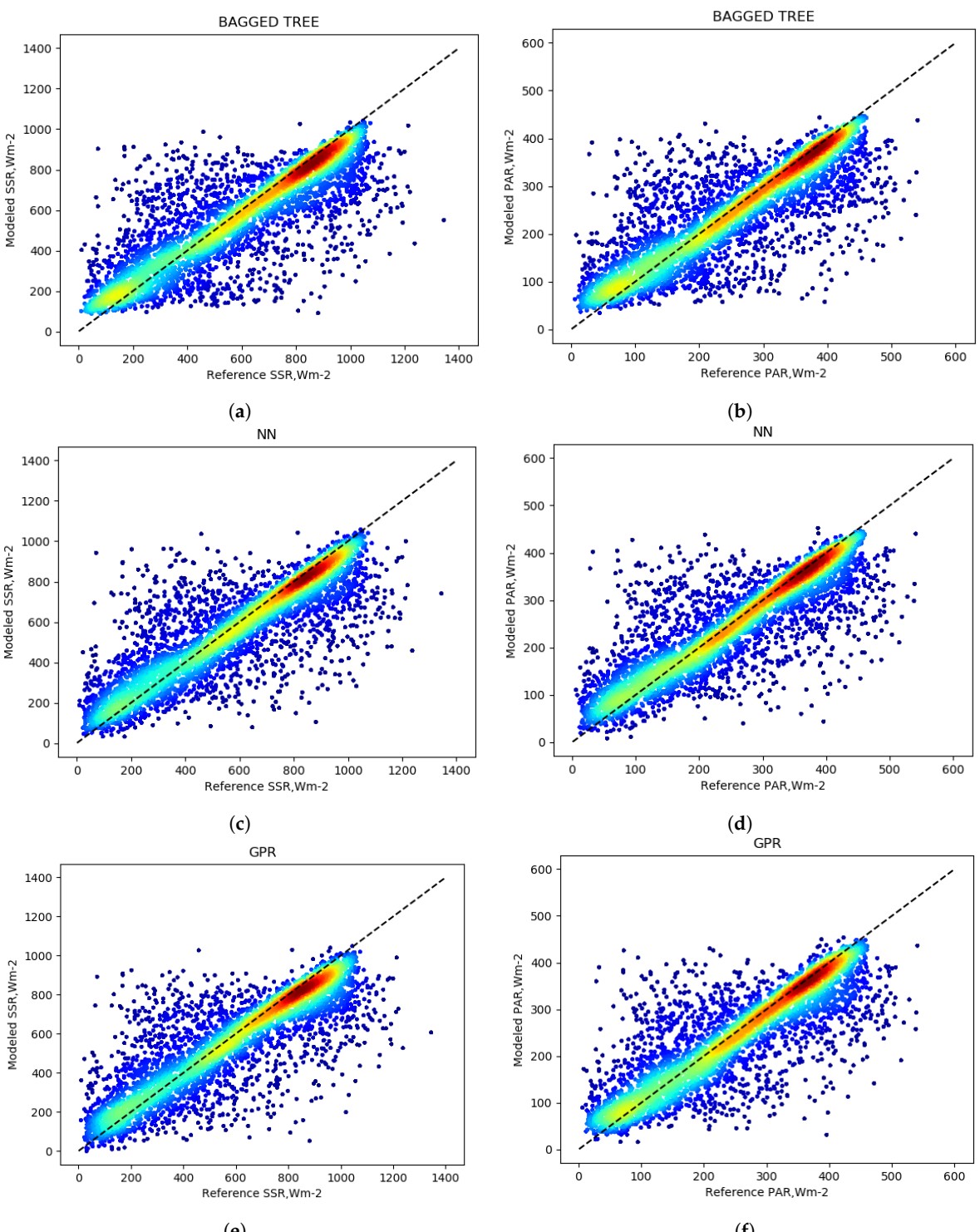

**Figure 3.** Validation Method 1 results as follows: (**a**) SSR BAGGED TREE. (**b**) PAR BAGGED TREE. (**c**) SSR NN. (**d**) PAR NN. (**e**) SSR GPR. (**f**) PAR GPR. All methods are in good agreement, but show some deviation from the 1:1 line, especially at low values of PAR or SSR. The GPR method best corrects this effect, but the overall spread of the modeled radiation at low values increases somewhat compared the NN and BAGGED TREE methods.

### 4.2. Time Series and Site Analysis

During training, two types of cross-validation were conducted to test the robustness of the methods. First, in Validation Method 2 of our analysis, we test the temporal stability of the model

methods. In Leave One Year Out Cross-Validation (LOYOCV), we iteratively train the model on only five of the six years and use the last year held out for cross-validation. In this way, we test the temporal robustness of the model methods, the statistics are given in Tables 3 and 4, and they show that the methods are temporally stable.

For the LOYOCV, we find that for these six years of data, the model is temporally stable, and there are no outlier years.

**Table 3.** Validation Method 2, Leave One Year Out Cross-Validation (LOYOCV) results for SSR.

| Method | $R^2$ | std | RMSE (W/m$^2$) | std (W/m$^2$) | Bias (W/m$^2$) | std (W/m$^2$) |
|---|---|---|---|---|---|---|
| RLR | 0.62 | 0.08 | 183 (30%) | 25 | 5 | 9 |
| LASSO | 0.65 | 0.07 | 182 (30%) | 18 | 51 | 12 |
| ELASTIC NET | 0.65 | 0.07 | 182 (30%) | 19 | 50 | 10 |
| DECISION TREE | 0.60 | 0.01 | 193 (32%) | 4 | −2 | 7 |
| BAGGED TREE | 0.77 | 0.01 | 140 (23%) | 6 | 0 | 6 |
| BOOSTED TREE | 0.73 | 0.02 | 151 (25%) | 7 | 1 | 7 |
| NN | 0.78 | 0.02 | 136 (22%) | 7 | 1 | 6 |
| KRR | 0.76 | 0.02 | 141 (23%) | 7 | 0 | 5 |
| GPR | 0.78 | 0.02 | 138 (23%) | 7 | 0 | 5 |

We find comparable results for the PAR LOYOCV. Keeping in mind that since PAR is approximately half of SSR, the RMSE and bias for the PAR are relatively the same as for SSR.

**Table 4.** Validation Method 2, Leave One Year Out Cross-Validation (LOYOCV) results for PAR.

| Method | $R^2$ | std | RMSE (W/m$^2$) | std (W/m$^2$) | Bias (W/m$^2$) | std (W/m$^2$) |
|---|---|---|---|---|---|---|
| RLR | 0.63 | 0.08 | 78 (30%) | 10 | 2 | 4 |
| LASSO | 0.65 | 0.06 | 77 (29%) | 8 | 22 | 5 |
| ELASTIC NET | 0.65 | 0.06 | 78 (29%) | 8 | 22 | 5 |
| DECISION TREE | 0.61 | 0.01 | 81 (31%) | 1 | −1 | 2 |
| BAGGED TREE | 0.77 | 0.02 | 60 (23%) | 2 | 0 | 2 |
| BOOSTED TREE | 0.73 | 0.02 | 64 (24%) | 3 | 0 | 2 |
| NN | 0.79 | 0.02 | 58 (22%) | 3 | 0 | 2 |
| KRR | 0.77 | 0.02 | 60 (23%) | 3 | 0 | 2 |
| GPR | 0.77 | 0.03 | 59 (22%) | 4 | -1 | 2 |

Second, in Validation Method 3, we tested the spatial stability, using the Leave One Station Out (LOSOCV) cross-validation approach, similar to Validation Method 2, we iteratively train on only six of the seven SURFRAD sites and use the site held out for cross-validation. The statistics are given in Tables 5 and 6 and discussed in the following sections.

**Table 5.** Validation Method 3, Leave One Site Out Cross-Validation (LOSOCV) results for SSR.

| Method | $R^2$ | std | RMSE (W/m$^2$) | std (W/m$^2$) | Bias (W/m$^2$) | std (W/m$^2$) |
|---|---|---|---|---|---|---|
| RLR | 0.60 | 0.09 | 182 (31%) | 33 | 7 | 17 |
| LASSO | 0.63 | 0.07 | 184 (31%) | 31 | 54 | 36 |
| ELASTIC NET | 0.63 | 0.07 | 183 (31%) | 31 | 52 | 36 |
| DECISION TREE | 0.51 | 0.10 | 214 (36%) | 27 | -19 | 55 |
| BAGGED TREE | 0.74 | 0.04 | 149 (25%) | 13 | -11 | 42 |
| BOOSTED TREE | 0.70 | 0.04 | 155 (26%) | 10 | -2 | 24 |
| NN | 0.76 | 0.04 | 139 (23%) | 12 | -8 | 30 |
| KRR | 0.73 | 0.04 | 146 (25%) | 15 | 8 | 13 |
| GPR | 0.75 | 0.04 | 141 (24%) | 15 | -2 | 15 |

For the PAR LOSOCV, the results are comparable to the SSR, as shown in Table 6.

**Table 6.** Validation Method 3 LOSOCV results for PAR.

| Method | $R^2$ | std | RMSE (W/m$^2$) | std (W/m$^2$) | Bias (W/m$^2$) | std (W/m$^2$) |
|---|---|---|---|---|---|---|
| RLR | 0.60 | 0.09 | 78 (31%) | 14 | 3 | 7 |
| LASSO | 0.63 | 0.07 | 78 (31%) | 12 | 21 | 16 |
| ELASTIC NET | 0.63 | 0.07 | 78 (30%) | 12 | 21 | 16 |
| DECISION TREE | 0.54 | 0.05 | 88 (34%) | 7 | −4 | 17 |
| BAGGED TREE | 0.74 | 0.04 | 64 (25%) | 5 | −5 | 18 |
| BOOSTED TREE | 0.71 | 0.04 | 67 (26%) | 4 | 0 | 16 |
| NN | 0.76 | 0.04 | 59 (23%) | 5 | −1 | 9 |
| KRR | 0.67 | 0.11 | 72 (28%) | 21 | −1 | 4 |
| GPR | 0.75 | 0.03 | 61 (24%) | 5 | 0 | 9 |

## 5. Discussion

The most accurate of the machine-learning methods were the bootstrap aggregated decision tree, the Neural Network, and the Gaussian Process Regression. Interestingly, not only were these the most accurate, they were also the fastest to run (approximately 30 and 45 min to train, respectively), which makes them practical methods to adopt for users looking to generate their own regional SSR or PAR estimates.

We find that regardless of the chosen method, the model is quite stable when we performed an iterative training and cross-validation through time and space. Among the SURFRAD sites, the Table Mountain site near Boulder, CO shows considerably different validation statistics from the other sites that skews the spatial cross-validation somewhat. Including more sites in the training and/or cross-validation may resolve this issue; however ground measurements from other networks in the United States do not have the same data quality or data record as the SURFRAD sites that were designed for long-term radiation monitoring [32].

The optical depth of the atmosphere, whether due to clouds or aerosols, still presents a challenge to this work. The thickness of clouds and aerosols is a major factor in how much radiation can reach the surface [46–48]. The aim of this work was to test if standard machine-learning methods could accurately estimate SSR and PAR without this a priori information, and we have shown that they can within 20% error. However, while machine learning can infer statistical relationships and make estimations based on those relationships, the missing information from the model will likely be seen in the comparison between these methods and other satellite estimates based on physical models.

We have reported our results as instantaneous estimates of SSR and PAR, while many other studies report 3-hourly estimates [15,49,50]. Zhang et al. [15] report RMSE of 12% for their 3-hourly estimates at the SURFRAD sites, while other estimates range from 14–24% at the same sites. The best comparison we can make is to the instantaneous SSR and PAR estimates from the new MODIS suite of products, MCD18. Wang et al. [16] report RMSE between 10–18% at the different SURFRAD sites.

## 6. Conclusions

In this work we tested nine machine-learning methods to model SSR and PAR using minimal input data from the MODIS instrument at 1 km spatial resolution in order to explore the ability of machine-learning-based, empirical model to estimate surface shortwave radiation (SSR) and photosynthetically active radiation (PAR) using input data from minimal sources to reduce error propagation and computational time. We found that the bootstrap aggregated decision tree (Bagged Tree), Gaussian Process Regression, and Neural Network yield the best results with minimal input and training data requirements. We report an $R^2$ of 0.77, 0.78, and 0.78 respectively, a bias of $0 \pm 6$, $0 \pm 6$, and $0 \pm 5$ W/m$^2$, and an RMSE of $140 \pm 7$, $135 \pm 8$, and $138 \pm 7$ W/m$^2$, respectively, for all-sky condition total surface shortwave radiation and viewing angles less than 55°.

Future studies should focus on several areas: (1) Adding more MODIS bands as inputs to the model. While the first 7 MODIS bands cover a large portion of the electromagnetic spectrum, some of

the other bands may be more sensitive to aerosols in the atmosphere that would limit radiation from reaching the surface. (2) Adding more training data to the model. Our work was aimed at finding the smallest reasonable training sample and the simplest reasonable model design, but more training samples, may improve the model assuming the differences in measurements and calibrations could be well handled. (3) More aggressive filtering of input and training data. Our intention was to include as much data as possible; however, starting from an idealized clear-sky model and building a more complex model to handle cloudy-sky cases could be one strategy to improve the model results. (4) Developing high temporal resolution direct and diffuse estimates of SSR and PAR as most current models estimate only total SSR or PAR.

**Author Contributions:** Conceptualization, M.G.L.B., T.H. and S.L.; Formal analysis, M.G.L.B.; Funding acquisition, S.L.; Investigation, M.G.L.B.; Methodology, M.G.L.B. and T.H.; Resources, S.L.; Software, S.S.; Supervision, S.S., T.H. and S.L.; Validation, M.G.L.B. and S.S.; Visualization, M.G.L.B.; Writing—Original Draft, M.G.L.B.; Writing—Review and Editing, M.G.L.B. and S.S. All authors have read and agreed to the published version of the manuscript.

**Acknowledgments:** This study was supported by NASA (Grant ID: 80NSSC18K0620). We thank the SURFRAD science teams for making their measurement data available and the Valencia group for making their machine-learning codes available.

**Conflicts of Interest:** The authors declare no conflict of interest.

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
