# Peer review of "Intercomparison of Machine-Learning Methods for Estimating Surface Shortwave and Photosynthetically Active Radiation"

_remotesensing, doi:10.3390/rs12030372_

Round 1
Reviewer 1 Report
The article is devoted to a comparative analysis of several models (methods) based on machine learning algorithms for predicting the values of SSR and PAR using a minor number of input data measured from a satellite (in particular, in seven channels of the MODIS spectrometer). The work contains new data and is of great interest to researchers using SSR and PAR in their work, especially if high temporal resolution and global coverage with such data is required.
Remarks:
1. The authors use two different terms to denote the same construct (which looks bad): in the heading of the article, in section 3 and some other places, “Method” is used, while the term “Model” is used throughout the text of the article?
2. In introduction: Due to the characterization of the disadvantages of the methods for solving the radiative transfer inverse problem using LUTs, which require, as the authors says, a lot of time to obtain data of interest (not calculating the LUTs themselves), I would like to see the typical errors of these methods and the computational time in comparison with the same values for the methods proposed in this paper (an error of 20% and the time to get one estimate are not given, it also does not say about the time it takes to prepare the data for training). The fact is that methods based on LUTs, as well as on the basis of radiation transfer models, which are widely used in problems of atmospheric correction of satellite data, can even obtain spectral radiation fluxes on the Earth's surface (instead of integral characteristics, SSR and PAR) with much more high precision.
3. Quote from the article:
62 estimating SSR and PAR from TOA measurements alone. Traditionally surface radiation estimates
63 are generated using radiative transfer models [17,18]. These models require information about the
64 top-of-atmosphere, the atmosphere, and the surface. Acquiring this ancillary data introduces sources
65 of potential error, requires heavy-duty computing resources, and is still time intensive [16]. Therefore,
66 it is desirable to have a model which can eliminate these extra sources of potential error, which can be
67 trained and executed quickly and efficiently, and still yield comparable results to existing methods.
Not all atmospheric correction models require additional information about the atmosphere and surface. As a rule, these are models using hyperspectral data, in particular, see [Katkovsky, L.V.; Martinov, A.O.; Siliuk, V.A.; Ivanov, D.A.; Kokhanovsky, A.A. Fast Atmospheric Correction Method for Hyperspectral Data. Remote Sens. 2018, 10, 1698.], which is also very fast, see also the analytical model for the Earth surface irradiation [https://rredc.nrel.gov/solar/pubs/spectral/model/section2.html#3]. At the same time, the data on the aerosol atmosphere (if necessary in the model) can be obtained from the nearest station of global AERONET network as well as meteorological data. In addition, the stations of the SURFRAD network that you use also provide the data necessary for radiation transfer models (atmospheric correction).
4. The data in table 1 require explanations:
• What is: 2002 - prs? 5-min. swath?
• How many points (multi-time data) were used for training and validation during the year?
• What were the temporal and spatial (for SURFRAD and MODIS) averaging of the measured data?
5. Quote from the article:
109 The SURFRAD instruments, mounted on platforms 1.5 to 2m off the ground, measure direct and
110 diffuse solar radiation, global solar radiation, UVB, PAR, aerosol optical depth, cloud cover, infrared
111 radiation as well as upwelling solar and infrared radiation, temperature, relative humidity, and wind.
Transcript UVB?
6. Quote from the article:
117 solar zenith angle. The uncertainty requirements for the SURFRAD instruments are 2-5% or 15W/m2
118 whichever is larger, to meet theWorld Climate Research Program specifications [31].
Is it possible to simultaneously introduce the measurements errors of TOA reflectance in MODIS channels so that the errors in used input data for modeling are understandable?
7. Quote from the article:
123 MODIS cloud condition (CM1-CM4) are used as inputs. The cloud condition is the only atmospheric
124 information contained in the model, aside from the TOA blue band, which can give some indication
125 of the clearness of the atmosphere. For each machine learning method, a separate model was trained
126 to predict SSR and PAR, as desired output.
It should be said that the values in all MODIS channels (not only in blue) depend in some extent on the state (parameters) of the atmosphere.
In this paragraph, the following expression is not clear, which requires explanation:
“For each machine learning method, a separate model was trained…”
What is meant by “separate model”? For example, it’s clear to me if it were said “separate data set” or “different optimal training sample size” (see line 172)!
Likewise, in lines 172 and 177, the expressions “Each model” are used, although, according to the preceding text, the expression “Each method” should apparently be (see remark 1)
8. In the following paragraph:
127 Prior to training, TOA reflectance from MOD021KM and MYD021KM and the overpass times
128 are extracted for pixels containing the location of surface sites. For each site, only one pixel is selected
129 and no spatial buffering is done at this time. This was done because, training was done prior to
130 reprojection and it is assumed that the effects of differing pixel sizes and the changing location of the
131 site within the pixel will average out with enough training data [16,32].
clarification is needed on the meaning of the terms in this context: “spatial buffering” and “reprojection”?
9. Quote from the article:
163 Data is filtered so that the training set contains only pixelswhose cloud flagmatches the expected
164 amount of ground measured radiation are used for training, however this a priori knowledge is not
165 applied to the validation data.
Does this mean that only cloudless pixels were used for training?
10. Fig. 2 is not very informative. In addition, the authors do not explain why the final values of R2 and RMSE are reached almost immediately, with a very little number of trees in bag?
11. There are absolutely no explanations for table 2 (or the authors believe that absolutely all readers should use, know and remember the notations of both the linear regression method itself and its coefficients)! Instead of reporting for the third time what the variables SZ, VZ, AZ, etc. mean (by the way, the error in the paragraph under table 2 is: twice AZ instead of VZ), would it be better if the authors gave definitions to the values in the table: SE, tStat, p-Value, and what do they mean with such amazing values, for example, 7.05E-219? Which of these values can we judge about the relative importance (influence) of the model input parameters?
12. In line 201, the abbreviation KRR is not defined (although it is clear which method it refers to).
13. Quote from the article:
215 radiation data, Robs is the mean of the ground measured radiation, and n are the number of data pairs.
Formulas (2) - (4) require explanations. It is necessary to explain what averaging is meant in this line for ground-based measurements: temporal, spatial (at one station or over all), by the number of measurements (how many were there?), To what number n is equal?
14. Quote from the article:
224 models. First, in Method 2 of our analysis, we test the temporal stability of the model. In Leave
225 One Year Out Cross Validation (LOYOCV), we iteratively train the models on only five of the six
226 years and use the last year held out for cross validation. In this way, we test the temporal robustness
Which of the years was left for validation or did each of the six years under review act in that role? Does the same question apply to 7 stations in method 3 (line 234)?
15. The methods LASSO and ELASTIC NET present in tables 4 and 5 were not described earlier in the article.
Author Response
MGLBROWN: Thank you for your input to our manuscript, we appreciate your help to improve our paper. We have done our best to address your concerns in the manuscript and responded to each of your remarks below.
Remarks:
The authors use two different terms to denote the same construct (which looks bad): in the heading of the article, in section 3 and some other places, “Method” is used, while the term “Model” is used throughout the text of the article?
MGLBROWN: Thank you for noticing this, we have corrected the text to better reflect that we are talking about machine learning methods that can be used in the SSR and PAR models.
In introduction: Due to the characterization of the disadvantages of the methods for solving the radiative transfer inverse problem using LUTs, which require, as the authors says, a lot of time to obtain data of interest (not calculating the LUTs themselves), I would like to see the typical errors of these methods and the computational time in comparison with the same values for the methods proposed in this paper (an error of 20% and the time to get one estimate are not given, it also does not say about the time it takes to prepare the data for training). The fact is that methods based on LUTs, as well as on the basis of radiation transfer models, which are widely used in problems of atmospheric correction of satellite data, can even obtain spectral radiation fluxes on the Earth's surface (instead of integral characteristics, SSR and PAR) with much more high precision.
MGLBROWN: Thank you for this comment. We have added the errors reported by several other papers, however, we note that most studies we have found report 3-hourly or daily SSR, and sometimes PAR, estimates whereas our estimates are instantaneous. Anecdotally from our colleagues we know that it can take upwards of a week to generate a LUT, but we struggled to find reported times for LUT generation.
Quote from the article:
62 estimating SSR and PAR from TOA measurements alone. Traditionally surface radiation estimates
63 are generated using radiative transfer models [17,18]. These models require information about the
64 top-of-atmosphere, the atmosphere, and the surface. Acquiring this ancillary data introduces sources
65 of potential error, requires heavy-duty computing resources, and is still time intensive [16]. Therefore,
66 it is desirable to have a model which can eliminate these extra sources of potential error, which can be
67 trained and executed quickly and efficiently, and still yield comparable results to existing methods.
Not all atmospheric correction models require additional information about the atmosphere and surface. As a rule, these are models using hyperspectral data, in particular, see [Katkovsky, L.V.; Martinov, A.O.; Siliuk, V.A.; Ivanov, D.A.; Kokhanovsky, A.A. Fast Atmospheric Correction Method for Hyperspectral Data. Remote Sens. 2018, 10, 1698.], which is also very fast, see also the analytical model for the Earth surface irradiation [https://rredc.nrel.gov/solar/pubs/spectral/model/section2.html#3]. At the same time, the data on the aerosol atmosphere (if necessary in the model) can be obtained from the nearest station of global AERONET network as well as meteorological data. In addition, the stations of the SURFRAD network that you use also provide the data necessary for radiation transfer models (atmospheric correction).
MGLBROWN: Thank you for this comment and reference. We have reworded this passage to emphasize that our goal for this study was to build a machine learning based, empirical model using *only* TOA measurements as input. We believe that other studies, Katkovsky et al as well as [Zhang, Y., He, T., Liang, S., Wang, D., & Yu, Y. (2018). Estimation of all-sky instantaneous surface incident shortwave radiation from Moderate Resolution Imaging Spectroradiometer data using optimization method. Remote sensing of environment, 209, 468-479.] do a much better job at leveraging atmospheric data and physical parameterization or machine learning into their more physical based approaches, however, our goal with this work was to test the power of machine learning methods in the simplest possible atmospheric model.
The data in table 1 require explanations:
What is: 2002 - prs? 5-min. swath?
How many points (multi-time data) were used for training and validation during the year?
What were the temporal and spatial (for SURFRAD and MODIS) averaging of the measured data?
MGLBROWN: Thank you for pointing this out to us. We have altered the table for clarity and added to the written explanation. We have added the number of data pairs to the description of the methods since it varies depending on which validation method is used.
Quote from the article:
109 The SURFRAD instruments, mounted on platforms 1.5 to 2m off the ground, measure direct and
110 diffuse solar radiation, global solar radiation, UVB, PAR, aerosol optical depth, cloud cover, infrared
111 radiation as well as upwelling solar and infrared radiation, temperature, relative humidity, and wind.
Transcript UVB?
MGLBROWN: Thank you for this comment, however we are unsure what is being asked here, therefore we have reworded the sentence to only include the measurements relevant to our experiments.
Quote from the article:
117 solar zenith angle. The uncertainty requirements for the SURFRAD instruments are 2-5% or 15W/m2
118 whichever is larger, to meet theWorld Climate Research Program specifications [31].
Is it possible to simultaneously introduce the measurements errors of TOA reflectance in MODIS channels so that the errors in used input data for modeling are understandable?
MGLBROWN:
Quote from the article:
123 MODIS cloud condition (CM1-CM4) are used as inputs. The cloud condition is the only atmospheric
124 information contained in the model, aside from the TOA blue band, which can give some indication
125 of the clearness of the atmosphere. For each machine learning method, a separate model was trained
126 to predict SSR and PAR, as desired output.
It should be said that the values in all MODIS channels (not only in blue) depend in some extent on the state (parameters) of the atmosphere.
In this paragraph, the following expression is not clear, which requires explanation:
“For each machine learning method, a separate model was trained…”
What is meant by “separate model”? For example, it’s clear to me if it were said “separate data set” or “different optimal training sample size” (see line 172)!
Likewise, in lines 172 and 177, the expressions “Each model” are used, although, according to the preceding text, the expression “Each method” should apparently be (see remark 1)
MGLBROWN: Thank you for this comment. Per remark 1, we have reworded the description of the methods used to distinguish between the machine learning methods used and the SSR and PAR models. We have also tried to emphasize in Section 3.2 that each method has a different optimal training sample size, and lastly, we have changed the wording that implies that only the MODIS blue band is affected by the atmosphere.
In the following paragraph:
127 Prior to training, TOA reflectance from MOD021KM and MYD021KM and the overpass times
128 are extracted for pixels containing the location of surface sites. For each site, only one pixel is selected
129 and no spatial buffering is done at this time. This was done because, training was done prior to
130 reprojection and it is assumed that the effects of differing pixel sizes and the changing location of the
131 site within the pixel will average out with enough training data [16,32].
clarification is needed on the meaning of the terms in this context: “spatial buffering” and “reprojection”?
MGLBROWN: Thank you for this comment, we have rephrased the text to indicate that we did not perform any spatial averaging, following the work of Zhang et al and Carter et al.
Quote from the article:
163 Data is filtered so that the training set contains only pixelswhose cloud flagmatches the expected
164 amount of ground measured radiation are used for training, however this a priori knowledge is not
165 applied to the validation data.
Does this mean that only cloudless pixels were used for training?
MGLBROWN: Thank you for this comment. To answer, no, both cloudy and clear pixels were used for training, but in the training data set we removed pixels where the cloud mask did not appear to match the ground state. We have reworded the text for clarification.
Fig. 2 is not very informative. In addition, the authors do not explain why the final values of R2 and RMSE are reached almost immediately, with a very little number of trees in bag?
MGLBROWN: Thank you for this comment, we have added an explanation as to why the final values of R2 and RMSE are reached so quickly with very few trees in the bag and eliminated the figure.
There are absolutely no explanations for table 2 (or the authors believe that absolutely all readers should use, know and remember the notations of both the linear regression method itself and its coefficients)! Instead of reporting for the third time what the variables SZ, VZ, AZ, etc. mean (by the way, the error in the paragraph under table 2 is: twice AZ instead of VZ), would it be better if the authors gave definitions to the values in the table: SE, tStat, p-Value, and what do they mean with such amazing values, for example, 7.05E-219? Which of these values can we judge about the relative importance (influence) of the model input parameters?
MGLBROWN: Thank you for this comment. We have replaced the table with a figure showing the variable coefficients of the model inputs and some descriptive text. We feel this better illustrates how the roles and relative importance of the model inputs.
In line 201, the abbreviation KRR is not defined (although it is clear which method it refers to).
MGLBROWN: Thank you for this comment, we have added the definitions of KRR and GPR to the text.
Quote from the article:
215 radiation data, Robs is the mean of the ground measured radiation, and n are the number of data pairs.
Formulas (2) - (4) require explanations. It is necessary to explain what averaging is meant in this line for ground-based measurements: temporal, spatial (at one station or over all), by the number of measurements (how many were there?), To what number n is equal?
Quote from the article:
224 models. First, in Method 2 of our analysis, we test the temporal stability of the model. In Leave
225 One Year Out Cross Validation (LOYOCV), we iteratively train the models on only five of the six
226 years and use the last year held out for cross validation. In this way, we test the temporal robustness
Which of the years was left for validation or did each of the six years under review act in that role? Does the same question apply to 7 stations in method 3 (line 234)?
MGLBROWN: Each of the six years under review acted in that role, same for the 7 stations, each station was left out for one iteration. We have clarified in the text that we are performing a type of k-fold cross validation where each of the 6 years was left out for one iteration, and likewise each of the sites was left out for one iteration of training and validation.
The methods LASSO and ELASTIC NET present in tables 4 and 5 were not described earlier in the article.
MGLBROWN: Thank you for this comment, we have added the definitions to the methods section.
Reviewer 2 Report
The manuscript compared machine learning models in estimating surface downward shortwave radiation from satellite radiation data and simplified cloud condition derived from satellite and reanalysis. The results showed that the selected machine learning models are reasonable approximators. Because the physical models based on radiative transfer simulations suffer from low quality input data and high computational demand, the machine learning models may be good tool for surface shortwave radiation estimation. There are minor concerns:
-- Only downward surface shortwave radiation was discussed, not upward or net. The authors didn't mention this in abstract or introduction.
-- Line 49: The first sentence discussed the low accuracy of satellite-based estimates, while the second sentence discussed ways to reduce computational demands. The logic should be made clearer here.
-- Line 66: Error cannot be eliminated but can be reduced.
-- Line 101: I couldn't understand this sentence: "The first sites were set up ... 2000s".
Author Response
MGLBROWN: Thank you for your input to our manuscript, we appreciate your time in reviewing and helping us improve our work. We have done our best to address your concerns and have responded directly to each point below.
-- Only downward surface shortwave radiation was discussed, not upward or net. The authors didn't mention this in abstract or introduction.
MGLBROWN: Thank you for this comment, we have added this specificity in the abstract and introduction
-- Line 49: The first sentence discussed the low accuracy of satellite-based estimates, while the second sentence discussed ways to reduce computational demands. The logic should be made clearer here.
MGLBROWN: Thank you for this comment. We have tried to better explain in the Introduction and Conclusion that our chosen methods and model design were both aimed at reducing uncertainty propagated through the model by multiple input data sources as well as test if it is possible to reduce computational time by using a relatively small training sample.
-- Line 66: Error cannot be eliminated but can be reduced.
MGLBROWN: Thank you for this comment, we have reworded the sentence.
-- Line 101: I couldn't understand this sentence: "The first sites were set up ... 2000s".
MGLBROWN: Thank you for this comment, we have reworded the sentence to clarify the establishment of the SURFRAD sites.